# Demographic Profiles, Etiological Spectrum, and Anatomical Locations of the Post-Hepatic Obstructive Jaundice in Adult Population in Hadhramout Region in Yemen

**DOI:** 10.3390/diseases12120333

**Published:** 2024-12-19

**Authors:** Sultan Abdulwadoud Alshoabi, Abdulkhaleq Ayedh Binnuhaid, Halah Fuad Muslem, Abdullgabbar M. Hamid, Fahad H. Alhazmi, Faisal A. Alrehily, Abdulaziz A. Qurashi, Osamah M. Abdulaal, Abdullah F. Alshamrani, Awatif M. Omer

**Affiliations:** 1Department of Diagnostic Radiology, College of Applied Medical Sciences, Taibah University, Al-Madinah Al-Munawwarah 41477, Saudi Arabia; 2Department of Specialized Surgery, Radiology Section, Faculty of Medicine, Hadhramout University, Hadhramaut, Yemen; 3Department of Internal Medicine, Dr. Suliman Al Habib Hospital Altakhasosi, Riyadh 12344, Saudi Arabia; 4Radiology Department, Rush University Medical Center, Chicago, IL 60612, USA

**Keywords:** obstructive jaundice, common bile duct (CBD), choledocholithiasis, CBD stricture, carcinoma of the pancreas, cholangiocarcinoma, Mirizzi syndrome

## Abstract

Background: Obstructive jaundice is a common health challenge in daily clinical practice caused by a heterogeneous group of benign and malignant conditions in or around extrahepatic bile ducts. This study aimed to investigate the causes of obstructive jaundice, analyze the age and sex distribution, and report the locations of obstruction. Methods: This was a retrospective study of electronic records of patients diagnosed with obstructive jaundice in the Hadhramout region in Yemen. Results: This study analyzed the data of 303 patients (mean age: 57 ± 17.99 years; range: 18–95 years); 60.40% (n = 183) were female, and 39.60% (n = 120) were male. The highest prevalence was found in middle-aged adults (n = 112, 36.96%), followed by the old (n = 101, 33.33%). Common bile duct (CBD) stones were the most common cause of obstructive jaundice (n = 175, 57.8%), followed by CBD stricture (n = 58, 19.1%), carcinoma of the head of the pancreas (n = 35, 11.6%), cholangiocarcinoma (n = 21, 6.9%), and external compression of the CBD (n = 2, 0.7%). CBD stones, cholangiocarcinoma, and ampulla of Vater masses were more prevalent in females (30.9%, 3.8%, and 2.2%, respectively) than in males (25.8%, 2.9%, and 1.7%, respectively). In contrast, CBD stricture and carcinoma of the pancreas were more frequent in males, occurring in 12.1% and 7.1% of male patients, respectively, compared to 7.9% and 4.9% in female patients. The primary obstruction site was the CBD (n = 254, 83.8%), followed by the head of the pancreas (n = 30, 9.9%), and the ampulla of Vater (n = 13, 4.3%). Conclusions: Obstructive jaundice predominantly affects middle-aged adults followed by the old-aged patients predominantly in females. The most common cause of obstructive jaundice was CBD stones, followed by CBD stricture, while carcinoma of the head of the pancreas was the most common malignant cause, followed by cholangiocarcinoma. Distal CBD is the most common anatomical location of obstructive jaundice.

## 1. Introduction:

Jaundice (hyperbilirubinemia) is a yellow discoloration of body tissues resulting from excessive accumulation of bilirubin due to increased production or impaired excretion [1]. The etiology of jaundice can be broadly divided into pre-hepatic, hepatic, and post-hepatic categories [2]. The pathogenesis of post-hepatic jaundice involves obstruction of the biliary drainage (obstructive jaundice) which blocks the passage of bile through the extrahepatic bile ducts outside the liver before entry into the duodenum via the ampulla of Vater [3,4]. Obstructive jaundice is caused by a heterogeneous group of benign and malignant conditions, which vary among individuals. The morbidity and mortality associated with obstructive jaundice depend on the cause of obstruction. These causes include choledocholithiasis, common bile duct (CBD) stricture, cholangiocarcinoma, carcinoma of the pancreas or gallbladder, periampullary carcinoma, and Mirizzi syndrome [5].

Transabdominal ultrasound imaging (TAUS) is the most commonly used screening modality, offering the advantages of being non-invasive, widely available, and inexpensive [6]. However, TAUS has limitations owing to its high operator dependence, which requires substantial knowledge and skills. Additionally, hepatobiliary artifacts may reduce the accuracy in CBD lesion diagnosis [7]. Endoscopic retrograde cholangiopancreatography (ERCP) provides rapid access to the bile ducts, facilitating optimal management of extra-hepatic bile duct stones and tumors [8]. Currently, ERCP is an established diagnostic and therapeutic method for CBD [9].

Obstructive jaundice is a common health challenge in daily practice and may lead to severe complications and adverse health outcomes. Management of this condition poses substantial diagnostic and therapeutic challenges for surgeons [10]. In the literature, there is a notable gap regarding the causes of obstructive jaundice in our region. This study aimed to investigate the most common causes of obstructive jaundice, analyze age and sex distributions, and identify the locations of obstruction in the extrahepatic bile ducts. This study reported common causes of obstructive jaundice in patients, including bile duct stone, bile duct stricture, cholangiocarcinoma, pancreatic carcinoma, mass at the ampulla of Vater, and external compression. We also identified the frequent anatomical locations of these causes in the extrahepatic bile ducts, such as the CBD, ampulla of Vater, head of the pancreas, and external compression. (Figure 1). To the best of our knowledge, this is the first study to explore these aspects of the Hadhramout region of the Republic of Yemen.

## 2. Materials and Methods

### 2.1. Study Design

This was a retrospective cross-sectional study of electronic records of patients diagnosed with obstructive jaundice between August 2017 and November 2023. This study was conducted at the Alsafwa Consultative Medical Center (ACMC) in Almukalla City, Hadhramout, Republic of Yemen. This study was approved by the Research Ethics Committee of ACMC in Almukalla City, Hadramout, Republic of Yemen (Approval Number: ACMC-11-23, 1 November 2023). All procedures were performed in accordance with the Declaration of Helsinki and all applicable standards and laws.

### 2.2. Sample Size

After ethical approval was obtained, this study involved 303 (n = 303) patients with clinically diagnosed obstructive jaundice who underwent TAUS and ERCP to identify the underlying cause.

### 2.3. Inclusion Criteria

This study included all of the patients clinically diagnosed with obstructive jaundice (n = 303) who underwent TAUS and ERCP to detect the cause of obstructive jaundice.

### 2.4. Exclusion Criteria

This study excluded the following categories:Patients with jaundice caused by pre-hepatic factors.Patients with obstructive jaundice caused by hepatic factors.Patients in whom the causes of obstructive jaundice were not definitively determined by ERCP (n = 88).

### 2.5. Data Acquisition

All patients underwent TAUS performed by a highly qualified radiologist with 13 years of postdoctoral experience in general ultrasound imaging. A Mindray DC30 ultrasound Machine with a deep curved transducer of 3.5 MHz was used to assess the CBD and biliary ducts in all patients. Following this, ERCP was conducted on all patients to confirm the diagnoses.

#### 2.5.1. Ultrasound Imaging Procedure

After fasting for at least six hours, patients underwent TAUS scanning. Scanning was performed in the supine position with or without the left lateral decubitus position to scan the biliary tree. After the gel application, the ultrasound probe was wrapped around the right subcostal region. The right upper quadrant region was scanned in the sagittal, coronal, and axial planes.

#### 2.5.2. ERCP Technique

After optimal preparation, written informed consent was obtained. According to the revised guidelines of the American Society of Gastrointestinal Endoscopy (ASGE) were reported by Jacob et al. [11], each patient involved in this study underwent ERCP for diagnostic and therapeutic purposes. The procedures were performed to ensure the consensus guidelines for perioperative management of patients undergoing ERCP reported by Azimaraghi et al. [12].

In the current study, the cause of obstructive jaundice was confirmed using ERCP in all patients. In our work, we present images of ERCP showing distal CBD stones obstructing it (Figure 2).

### 2.6. Statistical Analysis

The collected data were analyzed using the Statistical Package for Social Sciences (SPSS) version 25 (IBM, Armonk, NY, USA), and DATAtab Team (2023), (DATAtab: Online Statistics Calculator (DATAtab e.U. Graz, Austria)). This study reported common causes of obstructive jaundice in patients. Also, we compared demographic characteristics, such as sex and age groups, among the patients. Furthermore, this study analyzed the distribution of causes and locations according to patients’ sex and age groups. Categorical variables are expressed as frequencies and percentages, whereas continuous variables are presented as means with standard deviations. A chi-square test (*X*^2^) was used to compare the variables between the groups, with *p*-values considered significant when less than 0.05.

## 3. Results

### 3.1. Demographic Data

This study included 303 patients with an average age of 57 ± 17.99 years (range: 18–95). Of those, 60.40% (n = 183) were female, and 39.60% (n = 120) were male. The most commonly affected age group was the middle-aged adults (n = 112, 36.96%), followed by the old (n = 101, 33.33%) (Table 1). Boxplots show females were affected more than males across different age groups, with a notably smaller number of affected patients in the younger age group (<20 years) (Figure 3).

The boxplots show that females had a wider range of age than males in different causes of obstructed jaundice, except in masses where males had a wider age range and in external compression where there was no significant difference (Figure 3).

### 3.2. Etiological Data

The study findings indicated that CBD stones were the most common cause of obstructive jaundice (n = 175, 57.8%), followed by CBD stricture (n = 58, 19.1%), carcinoma of the pancreatic head (n = 35, 11.6%), cholangiocarcinoma (n = 21, 6.9%), and external compression of the CBD (n = 2, 0.7%) (Table 2).

The results indicated that CBD stones were the cause in 30.9% of females and 25.8% of males, cholangiocarcinoma was the cause in 3.8% of females and 2.9% of males, and masses in the ampulla of Vater were the cause in 2.2% of females and 1.7% of males. In contrast, CBD strictures were the cause in 12.1% of males compared to 7.9% of females, and pancreatic carcinoma was the cause in 7.1% of males against 4.9% of females (Figure 4). However, this study revealed no significant difference in the overall nature of the causes of obstructive jaundice between males and females (*p* = 0.336) (Table 3).

The current study demonstrated significant variation in the nature of the causes of obstructive jaundice across different age groups. CBD stones were most prevalent in middle-aged adults (n = 60, 34.29%), followed by the old (n = 52, 29.71%), and young adults (n = 46, 26.28%). Cholangiocarcinoma was more prevalent in old patients (n = 14, 66.66%); however, it was less common in the other age groups. Carcinoma of the head of the pancreas was predominantly observed in middle-aged adults (n = 19, 54.28%), followed by the old (n = 10, 28.57%); however, it was less common in the other age groups (*X*^2^ = 34.21, *p* = 0.025) (Table 4, Figure 5).

### 3.3. Anatomical Locations Data

Our results showed that the CBD was the most common location of obstructive jaundice caused by the CBD (n = 254, 83.8%), followed by the head of the pancreas (n = 30, 9.9%), the ampulla of Vater (n = 13, 4.3%), and external compression of the CBD (n = 6, 2.0%) (Table 5). Additionally, when comparing the different parts of the CBD, our results found that the distal CBD was the most common site of obstructive jaundice (n = 126, 41.6%), followed by the proximal part (n = 23, 7.6%), and the middle part (n = 4, 1.3%) (Table 6).

## 4. Discussion

An accurate diagnosis of the cause of post-hepatic obstructive jaundice is the cornerstone for optimal management planning at different ages. A higher prevalence was obtained in females (60.40%) compared to males (39.60%). This finding aligns with the results of several studies: Chalya et al. reported 1.3:1 female-to-male ratio [13], Bhutia et al. found a 1:0.35 female-to-male ratio [5], Gameraddin et al. reported 65.33% females to 34.66% males [14], and Shukla et al. noted a 2:3 male-to-female ratio [15]. The predominance of females in these cases has been attributed to a higher incidence of gallbladder stones in women, which is a known risk factor for choledocholithiasis and benign biliary obstruction [13,16]

Our results showed that the mean age of the patients affected by obstructive jaundice was 57 ± 17.99 years, consistent with the findings of Khan ZA, who reported a mean age of 56.68 ± 23.34 years [17]. The most commonly affected age groups were middle-aged adults (36.96%) and the old (33.33%), which partially aligns with Hasan’s study, indicating that half of the patients undergoing ERCP for obstructive jaundice were aged 41–60 [18]. Notably, the clinical presentation of choledocholithiasis may vary with age [19]. The current study included only patients presenting with obstructive jaundice, which might explain the age groups involved compared to other studies.

It is important to note that the causes of post-hepatic obstructive jaundice include a range of benign and malignant lesions. CBD stones are the most common benign cause of CBD obstruction [5]. These findings confirm our findings, in which CBD stones were identified as the most prevalent cause of obstructive jaundice (57.8%). In our study, CBD strictures accounted for 19.1% of cases, although their nature (i.e., benign or malignant) was not determined. Aljahdli reported that while the majority of CBD strictures are considered malignant, up to 30% can be benign, presenting a diagnostic and management challenge [20]. CBD strictures can arise from various etiologies including iatrogenic postcholecystectomy or liver transplantation; primary sclerosing cholangitis; chronic pancreatitis; autoimmune pancreatitis, autoimmune cholangitis, Mirizzi syndrome, tuberculous, viral, or parasitic infections; HIV cholangiopathy; ischemia; vasculitis; trauma; and post-radiation therapy. On the malignant side, strictures may result from pancreatic carcinoma, cholangiocarcinoma, or metastatic disease with external compression by lymph nodes [21].

Obstructive jaundice is a common clinical manifestation of pancreatic head and neck cancer [22]. Clinically, 90% of pancreatic neoplasms are malignant ductal adenocarcinomas. Bornman noted that two thirds of pancreatic carcinoma develop in the head of the pancreas, with many patients presenting with progressive obstructive jaundice [23]. In line with these observations, in the current study, we found that carcinoma of the head of the pancreas was the most common malignant cause of obstructive jaundice and the third most common cause of obstructive jaundice, accounting for 11.6% of all cases.

Cholangiocarcinoma encompasses a group of malignancies originating from different locations in the bile ducts, including the intrahepatic, hilar/perihilar (Klatskin tumor), and distal CBD [24]. Its peak incidence is observed during the sixth to the seventh decade of life, with a slight male predominance [25]. Contrary to these trends, cholangiocarcinoma emerged as the fourth most common cause of obstructive jaundice in our study and was more frequently observed in females. This deviation may be attributed to our specific focus on cholangiocarcinoma as a sole cause of obstructive jaundice.

The epidemiology, pathogenesis, and classification of biliary stones differ according to the stone location in intrahepatic and/or extrahepatic bile ducts. The primary bile duct stones are mostly brown-pigment (calcium bilirubin) stones and more common in East Asian countries than in Western countries. Gallbladder stones are primarily cholesterol or black pigment stones and common in western countries and Japan [26]. In our study, CBD stones were the cause of obstructive jaundice in 57.8% of the study population, followed by CBD stricture (19.1%), carcinoma of the pancreatic head (11.6%), cholangiocarcinoma (6.9%), and external compression of the CBD (0.7%). These results reflected significant variation from the results of a similar study in the Sikkim state of India where CBD stones were the cause of obstructive jaundice in 84.93% of all patients, followed by gallbladder carcinoma (6.85%), followed by CBD stricture (2.74%), and pancreatic carcinoma (1.37%), cholangiocarcinoma (1.37%), periampullary carcinoma (1.37%), and Mirizzi syndrome (1.37%) [5].

Finally, Mirizzi syndrome, a rare condition in which obstructive jaundice is caused by external compression of the CBD or common hepatic duct due to gallstones impacted in Hartman’s pouch, is a notable cause [27]. Other mechanical causes of external compression of the bile ducts include non-Hodgkin lymphoma [28], hepatic artery aneurysm [29], pancreaticoduodenal artery aneurysm [30], portal vein thrombosis [31], and diaphragmatic hernia [32]. Interestingly, a recent case report suggested that cocaine is a potential cause of biliary tree obstruction [33]. In our study, we found two cases of obstructive jaundice due to external compression of the CBD, one caused by an enlarged hilar lymph node, and the other caused by a liver mass. This highlights that, while some causes are well-established, others may emerge in future research.

## 5. Limitations

This study is limited by the inability to precisely determine the locations of obstructive jaundice causes in different parts of the CBD in 101 (33.3%) patients. This lack of specific location data was a drawback, particularly when comparing the locations of CBD stones between those within the CBD. Despite this, we found that the distal CBD was the most common location for CBD stones.

## 6. Conclusions

Post-hepatic obstructive jaundice predominantly affects old adults followed by old-aged patients, with a higher incidence in females. The most common cause of obstructive jaundice was CBD stones followed by CBD stricture, which can be benign or malignant. Among the malignant causes, carcinoma of the head of the pancreas was the most common, followed by cholangiocarcinoma. The most common location for obstructive jaundice was identified as the CBD, particularly its distal part.

## Figures and Tables

**Figure 1 diseases-12-00333-f001:**
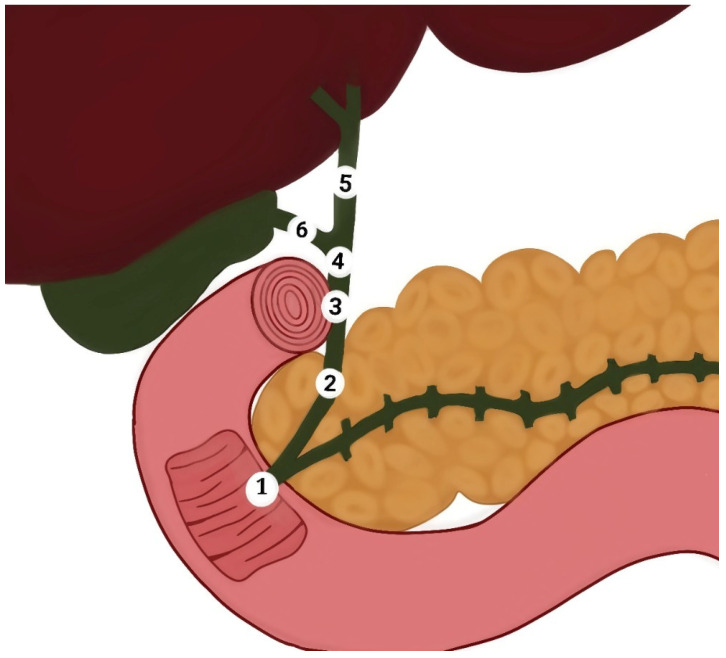
Diagram of the extrahepatic bile duct shows (1) ampulla of Vater, (2) retro pancreatic part of the common bile duct, (3) retro duodenal part of the common bile duct, (4) supraduodenal part of the common bile duct, (5) common hepatic duct, and (6) cystic duct.

**Figure 2 diseases-12-00333-f002:**
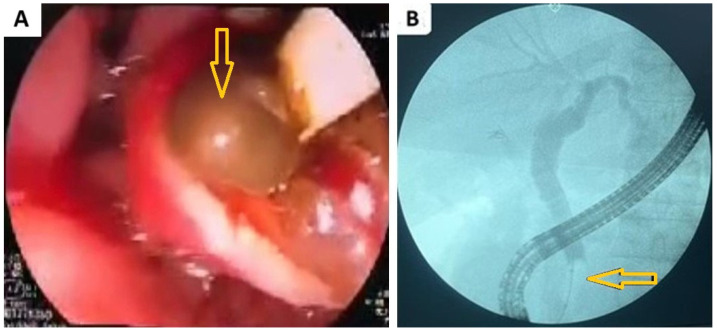
Shows (**A**) endoscopic retrograde cholangiopancreatography (ERCP) showing a stone (arrow) was impacted in the ampulla of Vater obstructing the bile ducts; (**B**) cholangiogram of ERCP showing multiple stones (arrow) in the distal common bile causing obstruction and dilatation.

**Figure 3 diseases-12-00333-f003:**
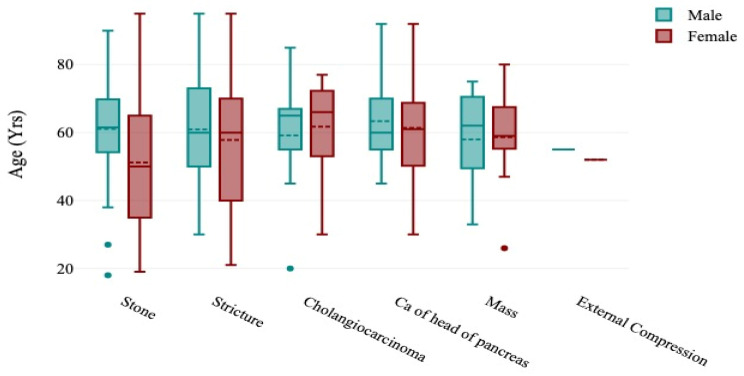
Boxplots showing the distribution of the causes of obstructive jaundice in the patients according to sex and age.

**Figure 4 diseases-12-00333-f004:**
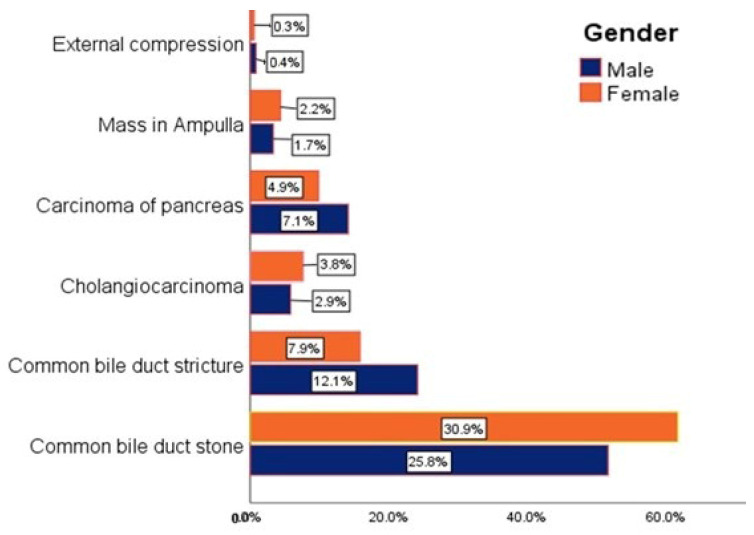
The distribution of causes of obstructive jaundice between males and females.

**Figure 5 diseases-12-00333-f005:**
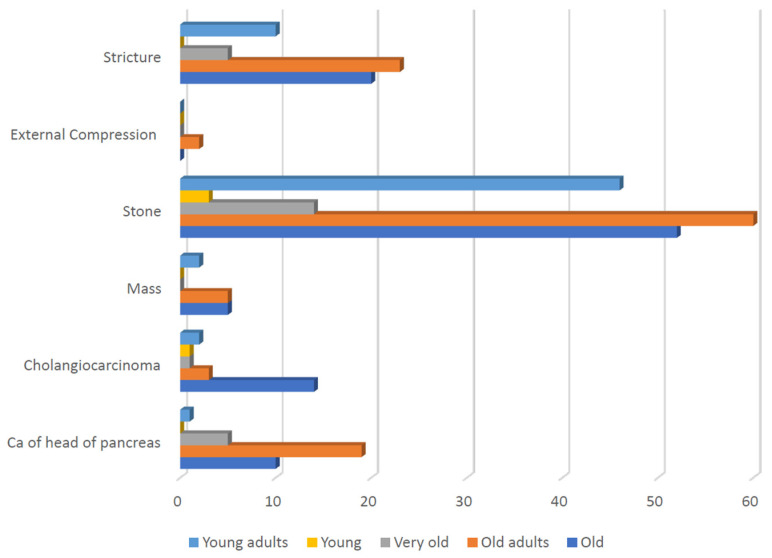
The difference in the distribution of the causes of obstructive jaundice among the different age groups.

**Table 1 diseases-12-00333-t001:** Sociodemographic features of the involved patients.

Variable	Categories	Number	Percentage
**Patient’s gender**	Male	120	39.60%
	Female	183	60.40%
**Patient’s age groups**	Young (<20 years)	4	1.32%
	Young adults (21–40 years)	61	20.13%
	Middle-aged adults (41–60 years)	**112**	**36.96%**
	Old (61–80 years)	**101**	**33.33%**
	Very old (>80 years)	25	8.25%

**Table 2 diseases-12-00333-t002:** The common causes of post-hepatic obstructive jaundice.

Causes of Obstructive Jaundice	Number	Percentage
Common bile duct stone	175	57.8%
Common bile duct stricture	58	19.1%
Cholangiocarcinoma	21	6.9%
Carcinoma of the pancreas	35	11.6%
Mass at the ampulla of Vater	12	4.0%
External compression	2	0.7%
Total	303	100%

**Table 3 diseases-12-00333-t003:** The common causes of obstructive jaundice in males and females.

Causes of Obstructive Jaundice	Male	Female	Total	*p*-Value
Common bile duct stone	62	113	175 (57.8%)	0.336
Common bile duct stricture	29	29	58 (19.1%)	
Cholangiocarcinoma	7	14	21 (6.9%)	
Carcinoma of the pancreas	17	18	35 (11.6%)	
Mass at the ampulla of Vater	4	8	12 (4.0%)	
Compression	1	1	2 (0.7%)	
Total	120	183	303 (100%)	

**Table 4 diseases-12-00333-t004:** The common causes of obstructive jaundice in different age groups.

Causes of Obstructive Jaundice	Young	Young Adults	Middle-Aged Adults	Old	Very Old	Total	*p*-Value
Common bile duct stone	3	46	60	52	14	175 (57.8%)	0.025
Common bile duct stricture	0	10	23	20	5	58 (19.1%)	
Cholangiocarcinoma	1	2	3	14	1	21 (6.9%)	
Carcinoma of the pancreas	0	1	19	10	5	35 (11.6%)	
Mass at the ampulla of Vater	0	2	5	5	0	12 (4.0%)	
External compression	0	0	2	0	0	2 (0.7%)	
Total	4	61	112	101	25	303 (100%)	

**Table 5 diseases-12-00333-t005:** The common locations of obstruction in the extrahepatic bile ducts.

Locations of Cause of Obstructive Jaundice	Number	Percentage
Common bile duct (CBD)	254	83.8%
Head of pancreas	30	9.9%
Ampulla of Vater	13	4.3%
Liver mass causing compression	6	2.0%
Total	303	100%

**Table 6 diseases-12-00333-t006:** The common locations of obstruction in the CBD and extrahepatic bile ducts.

Locations of Cause of Obstructive Jaundice	Number	Percentage
CBD stone (No determined part)	101	33.3%
CBD stone in the distal part	126	41.6%
CBD stone in the middle part	4	1.3%
CBD stone in the proximal part	23	7.6%
Head of pancreas	30	9.9%
Ampulla of Vater	13	4.3%
Liver mass causing compression	6	2.0%
Total	303	100%

CBD: common bile duct.

## Data Availability

Data are available from the corresponding author upon reasonable request.

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
