# Peer review of "Demographic Profiles, Etiological Spectrum, and Anatomical Locations of the Post-Hepatic Obstructive Jaundice in Adult Population in Hadhramout Region in Yemen"

_diseases, 2024, doi:10.3390/diseases12120333_

Round 1

Reviewer 1 Report (Previous Reviewer 3)

Comments and Suggestions for Authors

Thanks for response and limited changes in the manuscript

Author Response

We would like to thank you for your effort to revise our manuscript.

Reviewer 2 Report (Previous Reviewer 1)

Comments and Suggestions for Authors

This manuscript is a new version of an article that has already been evaluated previously and has improved considerably compared to its first version. However, the study's main weakness is that it is exclusively descriptive retrospective and has significant limitations.

The article would be improved if the authors were able to correct some errors in the text and answer some questions:

1- Page 3, line 98: Authors excluded patients in whom ERCP was not performed. These excluded patients were 88. Did these patients have different characteristics in terms of age, sex, or findings in the TUS compared to the 303 included? It would be good to clarify this in the text.

2.- Page 4, line 140 says “old adults,” and I think it should say “middle-age.”

3.- page 5, lines 144-145: the text says, “boxplots shows that females were affected more than males in different age groups with few affected patients in the group (<20 years)”, and reference is made to figure 3. However, figure 3 does not refer to the frequency in the different age groups but to the age of the patients according to the different causes of obstructed jaundice and sex. The authors should correct this sentence or remove the reference to Figure 3.

4.- Page 6, line 169: the text says, “cholangiocarcinoma was more prevalent in older patients” but should say “in old patients.”

5.- page 7, line 180: where it says “(n=154, 83.8%)”, it should perhaps say “(n=254, 83.8%)”

6.- page 7, line 183: where it says “(n=154, 83.8%)”, it should perhaps say “(n=126, 41.6%)”.

Author Response

Comments 1: [Page 3, line 98: Authors excluded patients in whom ERCP was not performed. These excluded patients were 88. Did these patients have different characteristics in terms of age, sex, or findings in the TUS compared to the 303 included? It would be good to clarify this in the text.]

Response 1: We apology, the excluded 88 patient were not included in analysis so that it is difficult to compare between their findings and the findings of the included patients.

Comments 2: [-Page 4, line 140 says “old adults,” and I think it should say “middle-age.”]

Response 2: Thank you for your accurate revision. This was unintentional mistake and we corrected.

Comments 3: [page 5, lines 144-145: the text says, “boxplots shows that females were affected more than males in different age groups with few affected patients in the group (<20 years)”, and reference is made to figure 3. However, figure 3 does not refer to the frequency in the different age groups but to the age of the patients according to the different causes of obstructed jaundice and sex. The authors should correct this sentence or remove the reference to Figure 3.]

Response 3: We agree with this comment. We have corrected the sentence as the following: Boxplots shows that females were had wider range of age than males in different causes of obstructed jaundice except in masses where males had wider age range and in external compression where no significant difference (Figure 3).

Comments 4: [Page 6, line 169: the text says, “cholangiocarcinoma was more prevalent in older patients” but should say “in old patients.”]

Response 4: We agree with this comment. This was corrected.

Comments 5: [page 7, line 180: where it says “(n=154, 83.8%)”, it should perhaps say “(n=254, 83.8%)”]

Response 5: Thank you for your accurate revision. This mistake was corrected.

Comments 6: [page 7, line 183: where it says “(n=154, 83.8%)”, it should perhaps say “(n=126, 41.6%)”.]

Response 5: Thank you for your accurate revision. This mistake was corrected.

Reviewer 3 Report (Previous Reviewer 2)

Comments and Suggestions for Authors

The paper is adequately and successfully revised and got ready for publication in its current form. No more issues detected. Well done.

Author Response

We would like to thank you for your effort to revise our manuscript.

Round 2

Reviewer 2 Report (Previous Reviewer 1)

Comments and Suggestions for Authors

The authors adequately responded to most of the observations made in the previous review. Unfortunately, the authors could not compare the characteristics of the 88 excluded patients with the findings of the included patients. This comparison could have made the results more robust. Nevertheless, the article can be published.

This manuscript is a resubmission of an earlier submission. The following is a list of the peer review reports and author responses from that submission.

Round 1

Reviewer 1 Report

Comments and Suggestions for Authors

I have read the article “Demographic Profiles, Ethological Spectrum, and Anatomical Locations of the Post-Hepatic Obstructive Jaundice in Adults in Hadhramout Region in Yenem.” It is a descriptive study of a series of 303 patients diagnosed with obstructive jaundice by TAUS and ERCP. The main problem of the study is that it is merely descriptive and does not add new knowledge about this pathology except for purely local data. The authors should point out the contribution of this work to the general knowledge, at least in the discussion. Therefore, the article should be rejected. 

Some comments below are: 

-The introduction does not mention endoscopic ultrasonography as a diagnostic method.

-In Material and Methods, the authors should expand the information on the geographical study area. Are there other hospitals besides the one participating in the study? What is the total population of the area? What percentage of this population does the hospital serve?

-Did the authors exclude patients who were diagnosed only by TAUS? How many?

-Did they not perform endoscopic ultrasonography in any case?

-What entities do the authors include in the expression “obstructive jaundice by pre-hepatic factors”? (line 96) 

-How can the authors explain that in 101 patients, the location of the CBD Stone was not determined?

-Could the authors describe whether the distribution by etiology changes in the different years studied?

Author Response

Comments 1: [The introduction does not mention endoscopic ultrasonography as a diagnostic method.]

Response 1: Our study was focused on TAUS which is the available imaging modality in Hadhramout region. Endoscopic ultrasonography is not available in Hadhramout region.

Comments 2: [-In Material and Methods, the authors should expand the information on the geographical study area. Are there other hospitals besides the one participating in the study? What is the total population of the area? What percentage of this population does the hospital serve?]

Response 2: The total population of Hadhramout governorate approximately 1.5 million. There are other hospitals in this region, however, the Alsafwa consultative medical center (ACMC) is the reference center of all these hospitals and ACMC has the only ERCP in Hadhramout which receives all cases need ERCP.

Comments 3: [Did the patients exclude patients who were diagnosed only by TAUS? How many?]

Response 3: [Yes, all authors with no available ERCP in addition to TAUS were excluded.

The No. was 88 patients] Thank you for pointing this out. We agree with this comment. Therefore, I/we have added the number of excluded patients due to no available ERCP (n=88) in page-3, excluded criteria, line-99.

Comments 4: [Did they not perform endoscopic ultrasonography in any case?]

Response 4: [No. Unfortunately, endoscopic ultrasonography is not available in Hadhramout region.]

Comments 5: [What entities do the authors include in the expression “obstructive jaundice by pre-hepatic factors”? (line 96)]

Response 5: We mean jaundice with prehepatic causes. This unintentional mistake was corrected. Thank you for pointing this out. Correction was done in page-3, exclusion criteria, line-96.

Comments 6: [How can the authors explain that in 101 patients, the location of the CBD Stone was not determined?]

Response 6: In the electronic records 101 patients, the cause was mentioned in the CBD, however, not determined in which part of the CBD. In the other patients, the causes were determined as proximal, middle, and distal parts of the CBD.

Comments 7: [Could the authors describe whether the distribution by etiology changes in the different years studied?]

Response 7: We apology. This needs recollection of the data to focus on the time of reports. This is difficult now.

Reviewer 2 Report

Comments and Suggestions for Authors

Authors studied the causes and characteristics of obstructive jaundice in patients from the Hadhramout region of Yemen in a retrospective work. The study clearly outlines its aim to investigate the causes, age/sex distribution, and locations of obstruction in patients with obstructive jaundice, providing a comprehensive look at this clinical issue. With 303 patients, the study offers a reasonably large sample for analyzing obstructive jaundice, which strengthens the reliability of the findings. However, few issues need to be addressed.

- The title should be revised as follows: "Demographic Profiles, Etiological Spectrum, and Anatomical Locations of the Post-Hepatic Obstructive Jaundice in Adult Population in Hadhramout Region in Yemen".

- Authors identify the causes and demographics of obstructive jaundice, but they do not address the clinical implications or potential changes in treatment strategies based on these findings. Including such a discussion would strengthen the abstract’s relevance to clinical practice.

- Although the study focuses on patients in the Hadhramout region, it does not address how generalizable these findings are to other regions or populations. Including a discussion on whether the causes and characteristics of obstructive jaundice might vary geographically could broaden the relevance of the study.

- Authors should compare the findings with those from other studies or regions, missing an opportunity to place the results in a broader global context. Adding this would help readers understand how common or unique the findings are to this specific population.

- P values must be provided in tables 3 and 4.

Author Response

1. Summary

2. Point-by-point response to Comments and Suggestions for Authors

Comments 1: [- The title should be revised as follows: "Demographic Profiles, Etiological Spectrum, and Anatomical Locations of the Post-Hepatic Obstructive Jaundice in Adult Population in Hadhramout Region in Yemen".]

Response 1: [This was done.] Thank you for pointing this out. We agree with this comment.

Comments 2: [Authors identify the causes and demographics of obstructive jaundice, but they do not address the clinical implications or potential changes in treatment strategies based on these findings. Including such a discussion would strengthen the abstract’s relevance to clinical practice.]

Response 2: This study focused on the causes and locations of obstructive jaundice in Hadhramout region which may change in prevalence from region to other. Treatment strategies are international procedures with no regional differences.

Comments 3: [Although the study focuses on patients in the Hadhramout region, it does not address how generalizable these findings are to other regions or populations. Including a discussion on whether the causes and characteristics of obstructive jaundice might vary geographically could broaden the relevance of the study.]

Response 3: [We add a paragraph to discuss the geographical variation in the causes of obstructive jaundice with references No. 26&27.] Thank you for pointing this out. We agree with this comment. Therefore, we have added the following paragraph and reference No. 26 “[The epidemiology, pathogenesis, and classification of biliary stones differ according to the stone location in intrahepatic and/or extrahepatic bile ducts. The primary bile duct stones are mostly brown-pigment (Calcium bilirubin) stones and more common in East Asian countries than in Western countries. Gallbladder stones are primarily cholesterol or black pigment stones and common in western countries and Japan [26].” in page-8, discussion, line 233-237.

Comments 4: [Authors should compare the findings with those from other studies or regions, missing an opportunity to place the results in a broader global context. Adding this would help readers understand how common or unique the findings are to this specific population.]

Response 4: [We add a paragraph to compare the geographical variation in the causes of obstructive jaundice in Hadhramout in comparison with other regions.] Thank you for pointing this out. We agree with this comment. Therefore, we have added the following sentences and reference No. 27 “In our study, CBD stone were the cause of obstructive jaundice in 57.8% of the study population, followed by CBD stricture (19.1%), carcinoma of the pancreatic head (11.6%), cholangiocarcinoma (6.9%), and external compression of the CBD (0.7%). These results reflected significantly variation from results of similar study in Sikkim state of India where CBD stones were the cause of obstructive jaundice in 84.93%, followed by Gallbladder carcinoma (6.85%), followed by CBD stricture (2.74%), and pancreatic carcinoma (1.37), Cholangiocarcinoma (1.37), periampullary carcinoma (1.37), and Mirizzi syndrome (1.37) of all patients [27].” in page-8, discussion, line 237-244.

Comments 5: [P values must be provided in tables 3.]

Response 5: Agree. We have, accordingly, added the p-value in tables 3&4 in page-6, results, inside the tables-3&4.

Reviewer 3 Report

Comments and Suggestions for Authors

I have the following comments

1. ERCP is not doen in all patients with extrahepatic biliary obstruction (EHBO), such as those with type 3 or 4 biliary block. Hence, the inclusion criteria shall be clinical plus USG evidence of EHBO.

2. Please provide the number of people with clinically suspected biliary obstruction were excluded due to non-availability of (i) USG and/or (ii) ERCP

3. If possible, please provide data on number of people with intrahepatic cholestasis evaluated during the study period

4. Provide information on clinical presentation of the participants ioncluded in this study

5. Proportion of participants who presented with cholangitis for different etiology

6. Table 1: Age categories shall be reduced to a maximum or two or three to make a sense. I will sugests for theose <40 and >40 years of age

7. Present the cause of obstruction in context of benigh and malignant etiology

8. Figure 3: Please place the box plots of male and female adjacent to each other for each etiology

9. The commonest cause was stione disease, hence please, if possible, provide the data on number and size of biliary stones

Thanks

Comments on the Quality of English Language

Moderate editing will be needed

Author Response

1. Summary

2. Point-by-point response to Comments and Suggestions for Authors

Comments 1: [ERCP is not done in all patients with extrahepatic biliary obstruction (EHBO), such as those with type 3 or 4 biliary 2block. Hence, the inclusion criteria shall be clinical plus SG evidence of EHBO]

Response 1: Our study was involved only the cases in which ERCP was done and diagnosis was confirmed. No cases without ERCP included in this study.

Comments 2: [Please provide the number of people with clinically suspected biliary obstruction were excluded due to non-availability of (i) USG and/or (ii) ERCP].

Response 2: [The No. was 88 patients] Thank you for pointing this out. We agree with this comment. Therefore, we have added the number of excluded patients due to no available ERCP (n=88) in page-3, excluded criteria, line-99. TAUS was available in all cases.

Comments 3: [If possible, please provide data on number of people with intrahepatic cholestasis evaluated during the study period]

Response 3: [We apology, this needs recollection of the data which is difficult.].

Comments 4: [Provide information on clinical presentation of the participants included in this study]

Response 4: [All cases were presented with jaundice. No other clinical features were collected.]

Comments 5: [Proportion of participants who presented with cholangitis for different etiology]

Response 5: This needs recollection of the data, however, this was not a focus in this study.

Comments 6: [Table 1: Age categories shall be reduced to a maximum or two or three to make a sense. I will suggest for those <40 and >40 years of age]

Response 6: In this study, we wanted to present the cause of obstructive jaundice in the four different age groups mentioned in chronological references (young, young adults, middle-age adults, and old) to assure diversity in prevalence of obstructive jaundice.

However, if it is necessary, we can change the categories into two groups.

Comments 7: [Present the cause of obstruction in context of benign and malignant etiology]

Response 7: This cannot be done because of some of the causes, such as stricture and external compression, may be benign or malignant.

Comments 8: [Figure 3: Please place the box plots of male and female adjacent to each other for each etiology].

Response 8: [This was done.] Thank you for pointing this out. We agree with this comment. Therefore, we have changed figure 3 in page-5 according to the comment.

Comments 9: [The commonest cause was stone disease, hence please, if possible, provide the data on number and size of biliary stones]

Response 9: This needs recollection of the data, however, this was not a focus in this study.

[In comment-9, the Reviewer requested data on number and size of biliary stones which was not a focus point in this study, and we need to recollect data for this which is difficult now.]